# Personal Experiences and Preferences for Weight-Management Services from Adults Living with Overweight and Obesity in the United Kingdom

**DOI:** 10.3390/nu16132016

**Published:** 2024-06-26

**Authors:** Eleanor Kyle, Aoibhin Kelly, Laura McGowan

**Affiliations:** 1Centre for Public Health, School of Medicine, Dentistry and Biomedical Sciences, Queen’s University Belfast, Belfast BT12 6BA, UK; ekyle10@qub.ac.uk (E.K.); laura.mcgowan@qub.ac.uk (L.M.); 2Institute for Global Food Security, School of Biological Sciences, Queen’s University Belfast, Belfast BT12 6BA, UK

**Keywords:** obesity, weight management, behavioural weight-management programme, public preferences

## Abstract

Evidence-based approaches for weight management in the United Kingdom are lacking. This study examined preferences for behavioural weight-management programmes amongst adults aged eighteen and over in Northern Ireland who had experience living with overweight (i.e., body mass index (BMI) ≥ 25 kg/m^2^). It involved the design and implementation of an online survey assessing previous experience with weight management and preferences for future weight-management programmes. A total of 94.7% of participants had previously engaged with weight-management services but many struggled to maintain weight loss. Older adults were more likely to be motivated to reduce their weight whilst younger adults were more likely be motivated to change their appearance. A focus on both wellbeing and weight-related outcomes was evident. Participants preferred programmes to be low-cost, delivered by a range of professionals by blended delivery, consisting of short (≤1 h) weekly sessions. These preferences highlighted important considerations for the components of future services to improve engagement and effectiveness.

## 1. Introduction

Obesity is a chronic progressive disease which is defined as “abnormal or excessive fat accumulation that may impair health” [1]. In 2016, the World Health Organization (WHO) reported more than 1.9 billion adults living with excess weight or obesity (defined as having a BMI ≥ 25 kg/m^2^ and BMI ≥ 30 kg/m^2^, respectively) [1]. Obesity increases the risk of developing comorbidities such as cardiovascular disease, diabetes, chronic kidney disease, and cancer, significantly increasing mortality [2].

The National Institute for Health and Care Excellence (NICE) guidelines in England (typically followed in Northern Ireland (NI)) historically outlined evidence-based weight-management strategies in a tiered-system [3,4]. These include universal health promotion and obesity prevention measures (Tier 1) aimed at the general population, and community-based behavioural interventions (Tier 2) for those with overweight/obesity (BMI ≥ 25 kg/m^2^). Tiers 3 and 4, now typically referred to as specialist weight-management services, encompass surgical, dietetic, pharmacological, and psychological obesity-management interventions for the management of obesity (BMI ≥ 30 kg/m^2^) including complex obesity (BMI of 35–40 kg/m^2^+), with accompanying psychological and physical comorbidities. The National Health Service (NHS) recommendations for community-based weight management (Tier 2) include multicomponent interventions targeting behavioural changes (i.e., diet and physical activity) [5]. Effective behavioural weight-management programmes (BWMPs) remain limited throughout the United Kingdom (UK), particularly in NI. Further understanding of preferences of people living with overweight and obesity (PLwO) is essential to improve the design, implementation, and reach of BMWPs. Gaining such knowledge is an asset for future programme development. With a new obesity strategy for NI currently under development following “A Fitter Future for All 2012–2022”, this is an opportune time to investigate the preferences of PLwO, with the hope that this can influence future weight-management service (WMS) provision.

This study aimed to:Evaluate previous engagement and experience with BWMPs and broader WMS.Determine public preferences regarding design of future BWMPs, including motivations and goals.Identify differences in preferences between various demographic characteristics, including age and gender.

## 2. Methods

### 2.1. Study Design

A brief, online, cross-sectional, quantitative questionnaire was developed using previous research and patient (or “personal” in NI) and public involvement (PPI) and hosted using Qualtrics (Computer Software) [2021, available online: https://www.qualtrics.com/core-xm/ (accessed on 14 June 2022)]. Four PLwO collaborated with the study team to improve design, wording, and content. Before release, the survey was piloted with ten participants for final refinements. Participation was voluntary and anonymous. The survey was distributed via social media and recruitment advertisements were shared with relevant organisations, community groups, and university circulars. Eligible participants were adults aged eighteen years or over, living in NI, who currently or previously had experienced living with excess weight (i.e., a BMI ≥ 25 kg/m^2^). Participants were required to have the ability to read and understand English and access the survey digitally. No exclusions were applied regarding any mental/physical health conditions. Ethical approval was granted: Faculty REC Reference Number: MHLS 21_154.

### 2.2. Study Development

Demographic information was collected, including age (in categories), gender, and self-reported height and weight data (to calculate BMI) with questions adapted from the Health Survey NI 2019/20 and NI Census 2021 [6,7]. Participant’s preferences for BWMPs including mode of delivery, cost, leader, session duration, goals, progress monitoring methods, motivations, and barriers were explored. Many questions were adapted from existing validated surveys and literature on public preferences for components of BWMPs [8,9,10]. 

### 2.3. Sample Size

The desired sample size was calculated based on approximately 65% of 1.49 million adults in Northern Ireland living with excess weight (BMI ≥ 25 kg/m^2^) [7]. Based on this, a desired confidence level of 95%, and 5% margin of error, we determined that *n* = 384 responses were required to obtain a representative sample. 

### 2.4. Statistical Analysis

Data were exported for analysis using IBM SPSS v26. Suspected bot responses and responses that were less than 70% complete were systematically removed before analysis, along with any entries that did not meet the inclusion criteria (e.g., participant did not have personal experience of living with excess weight, or they initiated the survey but did not consent to all statements in order to proceed). Descriptive statistics were used to characterise demographic characteristics. Chi^2^ tests/Fisher’s exact test cross-tabulation statistics were used for comparing categorical variables, and independent *t*-tests were used for comparing associations between two groups with continuous variables. Experiences of BMWPs, barriers to engaging with BMWPs, and preferences for future BWMPs were analysed mainly using descriptive statistics and Chi^2^ /Fisher’s exact test cross-tabulation statistics and independent *t*-tests/analysis of variance (ANOVA) testing. Statistical significance was set at a *p*-value of ≤0.05 for all analyses. 

## 3. Results and Discussion

### 3.1. Sociodemographic Characteristics

A total of 323 survey responses were recorded, *n* = 228 were valid for analysis, excluding ineligible and incomplete responses (i.e., those with over 30% missing data, as a cut-off of at least 70% progress was set as the threshold for completion of the survey) and possible spam/bot completions. For analysis, age was categorised into a younger age group (18–44-year-olds, *n* = 141), and an older age group (45+ years, *n* = 87). See Table 1 below for sample sociodemographic characteristics.

### 3.2. Engagement with WMS

Most participants (94.7%) previously tried to manage their weight independently (i.e., without professional help). Just 20.9% had sought help from WMS. Of those who accessed help, commercial programmes, e.g., Slimming World, were mostly reported. Many participants reporting successful initial weight loss also reported “regaining all weight lost”, or “regaining the weight lost and gaining more” (42.5%). A total of 38% of respondents were not aware of any WMS available to them.

Participants attempting to manage their weight independently more often than engaging with WMSs is consistent with the existing literature [11]. Of those who had engaged with WMSs, commercial BWMPs were most popular, potentially reflecting a lack of freely available BWMPs funded by the NHS in NI.

Results from current BWMPs often demonstrate initial weight loss followed by progressive regain for most people in the long term [12]. Only a minority of survey participants had been “successful” (maintained weight loss for 1 year), illustrating the ongoing challenge of long-term weight maintenance. This highlights the need for a better understanding of the factors that can facilitate long-term weight loss maintenance that current approaches often do not address adequately. Systematic review evidence suggests that many people are concerned about their ability to maintain weight loss in the long term following the cessation of their BWMP [13]. The incorporation of less-intensive ongoing support following the completion of a programme may be effective at helping participants maintain healthy behaviours.

Furthermore, a lack of awareness of available WMS, including BWMPs, was evident. Results from a large-scale UK survey show that significantly more people require help than those who access services [14]. Increasing public knowledge of available services and the benefit of engaging with services (similar to the focus on engaging with support for smoking cessation) may encourage earlier initiation of weight management, reducing the likelihood of developing weight-related comorbidities. 

### 3.3. Motivations and Barriers

Figure 1 below highlights reported motivations for engaging with WMS, with “improved health” being selected by 79% of participants (could select multiple options). “To reduce my weight” and “Self-esteem, mood and body image” were also popular (73% reported for both). More adults in the older age group selected “To reduce my weight” as a motivation than those in the younger age group (82.8% versus 67.4%). However, younger adults more often selected “appearance” compared with older adults (61.7% versus 47.1%).

“Time management” and “Lack of motivation” were highlighted as the most common barriers to accessing WMSs for 47% and 42% of respondents, respectively (Figure 1). Top rated barriers for the younger age group were “Time management” (51.8% compared with 39% of older adults) and “Cost” (44.7% compared with only 20.7% of older adults). There were also differences between the groups in terms of highlighting “Work commitments” (34.8% in younger and 29.9% in older adults) and “Self-consciousness or embarrassment” (29% in younger and 17.2% in older adults) as barriers. 

Improved health and other holistic outcomes (e.g., self-esteem, mood, and body image) were highly rated motivating factors for accessing BMWPs, beyond a focus on physical measurements. This reflects a welcome shift in attitudes, with gains in other aspects of health and wellbeing being prioritised, rather than a sole focus on weight-related outcomes. Reducing weight remains a common motive for PLwO; however, weight-neutral approaches may be justifiable alternatives alongside a consideration of broader mental and physical health benefits of engaging with BWMPs [15]. 

Common barriers to accessing services (including time management, lack of motivation, cost, and work commitments) differed slightly by age group, representing the need to tailor programmes for different ages/life stages. Psychological barriers were more common in younger people who also reported having a higher number of barriers overall, perhaps reflecting a higher number of priorities (i.e., young children, work, home life, etc.) which may restrict their capability and opportunity to engage with suitable behaviours. They also appear more likely to feel self-conscious or embarrassed about their weight. Similar attitudes are reflected throughout the literature, with lower self-esteem often associated with higher odds of an unhealthy weight [15]. Given this association, it may be beneficial to reframe the focus of future health promotion schemes from weight loss to self-care to encourage improved mental wellbeing (i.e., self-esteem) through healthy eating and exercise, which in turn may have a positive effect on weight loss.

### 3.4. Outcome Monitoring

The most preferred outcome measurements for BWMPs included quality of life, wellbeing, and mood (see Figure 2) (could choose more than one outcome). These were followed by health-related risk scores and physical activity changes. Regarding what participants believed programme outcomes should be based on, generally, body measurements and weight-related outcomes were less highly rated than health- and wellbeing-related measures. However, weight-related outcomes remained highly favoured as motivating factors. Weight management, supported in an evidence-based, sustainable way, remains extremely important for population health improvement [16]. 

### 3.5. Preferences for Components of BWMPs

Low cost appeared essential, with most people preferring programmes to be “free via the NHS” or “around GBP 5 weekly”. A combination of delivery mode (i.e., blend of face-to-face and remote) was favoured. A combination of professionals for delivery was also preferred (61.2%). Most participants preferred a session frequency of one per week and a duration of one hour or less was popular. “Ongoing support” was favoured over a set duration of time. 

With regard to programme cost, the evidence surrounding free vs. paid BMWP is mixed. One study comparing the effectiveness of different programmes found that the results of an NHS-ran intervention were comparable to that of a private-run programme over the same time period [17]. Both programmes met the UK DoH best practise guidelines of an average weight loss of at least 3%, with at least 30% of participants losing at least 5% of initial weight [17]. Whilst it is possible that the adherence to a free programme may be lower than that of a paid programme, this demonstrates clear evidence of the effectiveness of free programmes and it may be beneficial to increase their availability.

Interestingly, a combination approach to delivery (i.e., blended face-to-face and remote) was highly rated. Evidence suggests that participants value the convenience and anonymity of online delivery but face-to-face sessions are essential for developing a relationship between providers and participants [13]. A blended approach may be useful for achieving both. Weight-management apps, which have been shown to have moderate effects on weight and health outcomes, can utilise a blended approach and may also be effective as part of multicomponent delivery interventions [18].

A combination of professionals delivering programmes was also favoured, which complements evidence suggesting that interventions delivered by multidisciplinary teams show greater efficacy [19].

Participant preferences for a programme delivered once per week and for a duration of one hour or less is consistent with the literature suggesting that weekly sessions are important for maintaining motivation [13]. Participants of all ages also indicated favouring “ongoing support” rather than programmes of fixed duration. This is potentially an important approach considering the chronic, relapsing nature of obesity, providing further rationale for embedding ongoing support, particularly for the weight-maintenance phase [13]. 

### 3.6. Limitations

Limitations of this study relate to the survey sample being recruited opportunistically online, and consequently not fully representing the NI population in terms of demographic characteristics. Women were overrepresented, there were more younger adults versus older adults, and little variation in ethnicity (although this is representative of the NI population). The sample were also mostly employed fulltime and had a university degree, increasing the likelihood of a higher socioeconomic status. This may have unduly influenced results in terms of WMS preferences, in terms of costs and ability to participate in programmes face-to-face/online, etc., and should be noted with some caution. 

Additionally, a priori sample size was not achieved (aim *n* = 384), which may have impacted statistical testing. The age groups used for analysis do not necessarily accord with the wider literature and there may also be implications of having relied on IT/computer literacy and exclusion of at-risk groups. 

### 3.7. Conclusion and Recommendations for Future

This study revealed limited awareness of WMS, particularly BWMPs, among PLwO, highlighting the need to raise awareness and introduce a clear pathway of evidence-based support for PLwO, particularly in NI. A lack of awareness of what publicly funded WMSs are available was evident. Rates of weight loss maintenance/regain suggested that longer-term support should be prioritised to improve long-term results, and broader health and wellbeing goals should be incorporated into a traditional outcomes-based approach. Barriers and motivations for engaging in WMSs varied by age group. Participant’s preferences highlighted important considerations for future BWMPs to include low cost and ongoing support, delivered by a combination of professionals using blended online/face-to-face approaches with short session lengths. Outcomes of value to participants were both health-related and weight-related, with improvement to health and wellbeing holistically being as important as reducing weight. This was also reflected in the preferences for goal setting/monitoring of progress in future BWMPs. These findings should be taken into consideration, and future BWMPs should explore approaches to improve adherence and engagement by sustaining motivation, supporting healthy behaviours, and appealing to individual interests. Further research is required to explore weight-management preferences of a wider representative sample of the UK population, with consideration for underrepresented groups, including men and those from across the age and socioeconomic spectrum.

## Figures and Tables

**Figure 1 nutrients-16-02016-f001:**
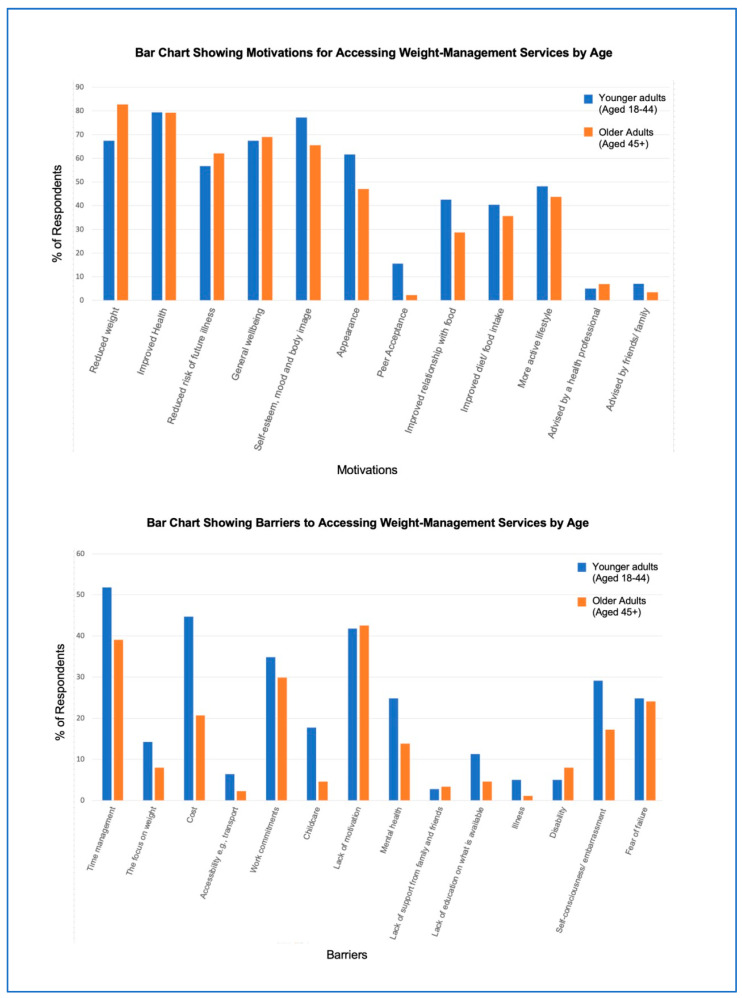
Cluster bar charts highlighting the motivations for, and barriers to, accessing weight-management programmes across different age groups.

**Figure 2 nutrients-16-02016-f002:**
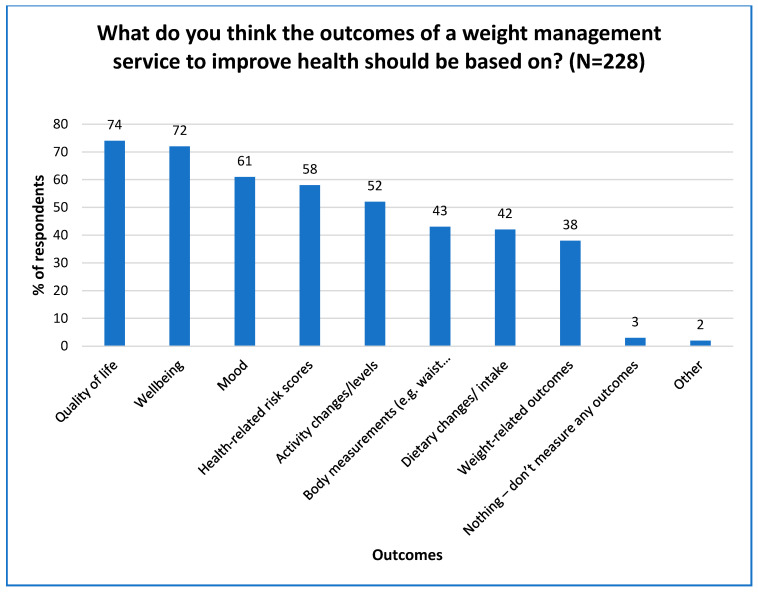
Bar chart showing behavioural weight-management programme outcomes that should be prioritised for health improvement (multiple options could be selected) (*n* = 228).

**Table 1 nutrients-16-02016-t001:** Survey participant characteristics (*n* = 228).

Characteristic	N (%)	Younger Adults (Aged 18–44) N = 141	Older Adults (Aged 45+) N = 87
**Gender**	**N = 228**	**N = 141**	**N = 87**
*Male* *Female*	47 (20.6) 181 (79.4)	33 (23.4) 108 (76.5)	14 (16) 73 (83.9)
**Age**	**N = 228**		
*Younger adults (18–44 years)* *Older Adults (45+ years)*	141 (61.8) 87 (38.2)	---------------	--------------------
**Self-reported BMI ^‡^**	**N = 201**	**N = 127**	**N = 74**
*Healthy weight* (*≤24.99* kg/m^2^) *Overweight* (*25–29.99* kg/m^2^) *Living with obesity* (*≥30* kg/m^2^)	31 (15.4) 74 (36.8) 96 (47.8)	23 (18.1) 45 (35.4) 59 (46.5)	8 (10.8) 29 (39.2) 37 (50)
**Level of Education***	**N = 228**	**N = 141**	**N = 87**
*Compulsory education (GCSE/equivalent)* *A Level/ AS Level or equivalent* *University degree (varying levels)* *Other*	10 (4.4) 39 (17.1) 155 (67.9) 24 (10.5)	3 (2.1) 24 (17) 103 (73) 11 (7.7)	7 (8) 15 (17.2) 52 (59.8) 13 (14.9)
**Employment Status ^†^**	**N = 228**	**N = 141**	**N = 87**
*Employed full time/part-time* *Unemployed* *In full-time education (i.e., a student)* *Retired* *Other*	189 (82.9) 1 (0.4) 18 (7.9) 11 (4.8) 9 (4)	120 (85.1) 1 (0.7) 17 (12.1) 0 (0) 3 (2.1)	69 (79.3) 0 (0) 1 (1.1) 11 (12.6) 6 (6.9)
**Area of residence**	**N = 227**	**N = 141**	**N = 86**
*Urban* *Rural* *Prefer not to say*	148 (65.2) 76 (33.5) 3 (1.3)	92 (65.2) 46 (32.6) 3 (2.1)	56 (65.1) 30 (34.9) 0 (0)
**Dependents living at home (<18 years old) ***	**N = 227**	**N = 141**	**N = 86**
*Yes* *No*	77 (33.9) 150 (66.1)	55 (39) 86 (61)	22 (25.6) 64 (74.4)
**Current Life Stage ^†^**	**N = 225**	**N = 141**	**N = 84**
*Not planning on starting a family* *Would like to have children in the future* *Have a young family* *Have older child/children* *Not sure* *Prefer not to say* *Other* ^§^	37 (16.4) 58 (25.8) 63 (28) 48 (21.3) 8 (3.6) 3 (1.3) 8 (3.6)	22 (15.6) 58 (41.1) 54 (38.3) 0 (0) 7 (5) 0 (0) 0 (0)	15 (17.9) 0 (0) 9 (10.7) 48 (57.1) 1 (1.2) 3 (3.6) 8 (9.5)
**Self-perceived Weight Status ***	**N = 227**	**N = 141**	**N = 86**
*Healthy weight* *Overweight* *Very overweight* *(* healthy weight includes n = 2 uw)*	43 (18.9) 127 (55.9) 57 (25.1)	36 (25.5) 71 (50.4) 34 (24.1)	7 (8.1) 56 (65.1) 23 (26.7)
**Pre-existing condition that you feel may impact weight** ** ^†||^ ** **.**	**N = 227**	**N = 141**	**N = 86**
*Yes* *No* *Prefer not to say*	71 (31.3) 148 (65.2) 8 (3.5)	35 (24.8) 101 (71.6) 5 (3.5)	36 (41.9) 47 (54.7) 3 (3.5)

* Indicates *p* < 0.05 as occurred from chi^2^ test of association between age categories. ^†^ Indicates *p* < 0.05 as occurred from Fisher’s exact test of association between age categories. ^‡^ Calculated from self-reported height and weight data. ^§^ Examples of “Other” responses included “too old to have children” and “do not wish to have children”. ^||^ Conditions included depression, anxiety, stress, other mental health issues, eating disorders, fibromyalgia, arthritis, asthma, chronic pain, hypothyroidism, multiple sclerosis, PCOS, type 1 diabetes, autoimmune disorders, and knee/ankle injuries.

## Data Availability

The raw data supporting the conclusions of this article will be made available by the authors on request.

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
