# Peer review of "Personal Experiences and Preferences for Weight-Management Services from Adults Living with Overweight and Obesity in the United Kingdom"

_nutrients, 2024, doi:10.3390/nu16132016_

Round 1

Reviewer 1 Report

Comments and Suggestions for Authors

Overall this was an interesting survey that offered some insights into preferences for weight management services in Northern Ireland that could potentially be extrapolated to other settings, though I note some limitations in terms of the representation in the sample which I have highlighted below. Overall there were few issues with this paper however there were some areas that I believe need slightly more explanation or justification.

Introduction

While tiers of care or described in the second paragraph of the introduction, as an international reader it would be useful to know which weight range (if that is how they are designed) each of these tiers are designed to target. For example, is tier one for the general population, or for those who have overweight? Who is Tier 4 targeted to? Those with severe obesity? What tier is behavioural management? Or does it span across more than one tier?

Methods

It is good to see involvement of PPI (people will lived experience) involved in development of the survey, with some pilot testing.

It will be useful to have the survey attached as an appendix to the paper (supplementary online material), As it takes detailed reading of the results to ascertain exactly what was asked.

While the analysis strategy is appropriately described, there is no detail on how missing data was dealt with.

Results

Following on from this we are told in the results that 323 survey responses were recorded with 228 valid for analysis. Does this mean that there was a case wise deletion used for missing data? This may not be the most appropriate approach given that the calculated desired sample size was 384. More detail on the nature of the data missing would be useful. Were there not some responses that could still be included despite being incomplete.

Limitations large proportion of female participants and knows the university degree, employed full time.

Were any questions asked about weight maintenance? What questions were asked about cost? What items were asked about self-consciousness or embarrassment? Were any questions asked about stigma or weight bias?

Some discussion has crept into the results. For example in section 3.2 engaging with weight management services there is discussion around consistency with existing literature (paragraph 2), results from current behavioural weight loss programmes and maintenance of weight loss following this participation (paragraph 3), and discussion about results from a large scale UK survey (paragraph 4). However given this is a brief communication perhaps this is intentional as I'm unable to see a separate discussion section and I'm happy to concur with the editors on where this information is situated in the manuscript.

The conclusion in section 3.3 about shifting the focus of programmes to self care to improve mental health being is an excellent suggestion appropriate given the high prevalence of weight stigma across many settings. This is likely to be a good approach in some settings, though I do note that this sample is highly educated and mostly female therefore these results could not be extrapolated across broader populations such as those of lower education and men, who may value holistic programmes less.

While the over representation of females have been addressed in the limitation section (3.6), there is no discussion on how this would likely have influenced those results nor any mention of the highly educated nature of the sample which is an omission that should be corrected. As well hyper portion of the sample are employed full time which again has a bearing on the response to the cost questions for example as well as ability to be able to participate in things face to face. This should also be addressed in limitations.

Minor typographical error:

Introduction: World Health Organisation is spelt ‘Organization’

Reviewer 2 Report

Comments and Suggestions for Authors

Thank you for involving me in the review of this manuscript, which deals with evidence-based approaches to weight management in the UK.

Some comments:

Line 55. Was the questionnaire used validated on the sample population together study?

The authors state that participation was voluntary. Was there a population selection criterion? age range?

Calculating the statistical power of salpe size adds strength to the study

Was the study approved by an ethics committee?

Are the conclusions consistent with the results
